# Synthesis and Characterization of Coordination Compound [Eu(µ_2_-OC_2_H_5_)(btfa)(NO_3_)(phen)]_2_phen with High Luminescence Efficiency

**DOI:** 10.3390/nano12162788

**Published:** 2022-08-14

**Authors:** Ion P. Culeac, Victor I. Verlan, Olga T. Bordian, Vera E. Zubareva, Mihail S. Iovu, Ion I. Bulhac, Nichita A. Siminel, Anatolii V. Siminel, Geanina Mihai, Marius Enachescu

**Affiliations:** 1Institute of Applied Physics, MD-2028 Chisinau, Moldova; 2Institute of Chemistry, MD-2028 Chisinau, Moldova; 3Center for Surface Science and Nanotechnology, University Politehnica of Bucharest, 060042 Bucharest, Romania; 4S.C. NanoPRO START MC S.R.L., 110310 Pitesti, Romania; 5Academy of Romanian Scientists, 550044 Bucharest, Romania

**Keywords:** europium(III), dinuclear coordination compound, blue-light excitable, photoluminescence, quantum yield

## Abstract

A high-luminescent, blue-light excitable europium(III) coordination complex, [Eu(µ_2_-OC_2_H_5_)(btfa)(NO_3_)(phen)]_2_phen (**1**) {btfa = benzoyl trifluoroacetone, phen = 1,10-phenantroline}, has been synthesized and investigated. The complex was characterized by infrared (IR) and *photoluminescence* (PL) spectroscopy. The PL emission spectra of powder samples registered in a range of 10.7–300 K exhibit characteristic metal-centered luminescence bands, assigned to internal radiative transitions of the Eu^3+^ ion, ^5^D_1_→^7^F_j_ and ^5^D_0_→^7^F_j_ (*j* = 0–4). The high-resolution spectrum of the transition ^5^D_0_→^7^F_0_ shows that it consists of two narrow components, separated by 0.96 meV, which indicates the presence in the matrix of two different sites of the Eu^3+^ ion. The splitting pattern of ^5^D_0_→^7^F_j_ (*j* = 0–4) transitions indicates that europium ions are located in a low-symmetry environment. The absolute quantum yield and the sensitization efficiency were determined to be 49.2% and 89.3%, respectively. The complex can be excited with low-cost lasers at around 405 nm and is attractive for potential applications in optoelectronics and biochemistry.

## 1. Introduction

Coordination compounds of trivalent europium ion Eu^3+^, with strong absorption in the near-UV and UV region and high photoluminescence quantum yield in the visible region, are extensively studied [1,2,3,4] for various applications in optoelectronics, medicine, and biology [5,6,7,8]. Specifically, important drivers for extensive research efforts arise from the needs for sensing applications [9,10], tissue and cell imaging [11,12], drug delivery monitoring, luminescent probes for optical imaging, or X-ray computer tomography [13,14] in biomedical assays, bio-sensors, etc. [15,16,17]. Because the europium ions are characterized by a simple structure of the ^2S + 1^L_j_ multiplets with non-degenerate ^5^D_0_ and ^7^F_0_ levels, they are used as spectroscopic probes to acquire information about the symmetry at the Eu^3+^ site for the local environment around the Eu^3+^ ion [18,19].

In the case of lanthanide ions in a free state, the 4f-4f transitions are forbidden by the Laporte rule [18,19,20]. Consequently, due to a small absorption cross-section of the Eu^3+^, direct 4f excitation is very weak, and Eu^3+^ ions cannot be efficiently excited directly by the excitation light. Instead, the high luminescence efficiency of Eu(III) coordination compounds is determined by the “antenna effect” and the energy transfer from the matrix to the Eu^3+^ ion. In order to increase the efficiency of lanthanide ion excitation, they are usually incorporated into the matrix of organic or inorganic compounds, which plays the role of a sensitizer [3,16,20]. In this case, the ligands of the host matrix absorb the excitation light and transfer the excitation energy to energy levels of the Eu^3+^ ion, from which radiating excited levels can be populated.

Considerable efforts in the investigation of lanthanide materials are devoted to new multinuclear lanthanide complexes [21,22,23] and blue-light excitable compounds [11,24,25] with a great potential for development towards the needs of medicine, biochemistry, quantum storage devices, etc. In the present communication, we report the preparation, IR characterization, and preliminary photoluminescence properties of a novel, blue-light-excitable, dinuclear europium(III)-based coordination compound with high *emission* quantum yield [Eu(µ_2_-OC_2_H_5_)(btfa)(NO_3_)(phen)]_2_phen (**1**) {btfa = benzoyl trifluoracetone, phen = 1,10-phenantroline}.

## 2. Experimental

### 2.1. Chemicals and Materials

All chemicals and reagents were obtained from commercial sources and used without further purification. High purity starting reagents were obtained from Sigma-Aldrich (St. Louis, MO, USA) and used as received: Europium hexahydrate nitrate (purish; 99.9%), 1,10-phenanthroline (p.a.; ≥99%), benzoyl trifluoroacetone (p.a.; 99%), and sodium hydroxide (pure; ≥98%). The complex [Eu(µ_2_-OC_2_H_5_)(NO)_3_(phen)]_2_phen was synthesized as described elsewhere [26].

The mixture of benzoyl trifluoracetone 0.324 g (15 mmol) and 1,10-phenanthroline 0.090 g (5.5 mmol) was dissolved in 12 mL of ethanol (solution 1) under heating at 60 °C, while 0.223 g (5 mmol) of europium hexahydrate nitrate was dissolved in a mixture of 1 mL of ethanol and 2 mL of water (solution 2). Warm solution 1 was added dropwise to solution 2 under continuous stirring, and then 1.5 mL of the sodium hydroxide solution (1 N) was added to the mixture. A white polycrystalline solid precipitated and was filtered off, washed with ethanol, and then with ether. Finally, it was air-dried to give a white powder.

Yield: 0.29 g (38.98%).

Anal. Calcd for Eu_2_C_60_H_46_F_6_N_8_O_12_: C—48.40; H—3.11; N—7.53; Eu—20.41.

Found: C—48.61; H—2.93; N—7.24; Eu—20.03 (determined from residue).

The prepared compound is stable in air over a long period of time (Appendix A, Appendix A), it is soluble in ethanol, methanol, ether, dimethylformamide, and dimethylsulfoxide, however, it is insoluble in water. The proposed molecular structure of the complex [Eu(µ_2_-OC_2_H_5_)(btfa)(NO_3_)(phen)]_2_phen is illustrated in Figure 1.

### 2.2. Methods for Characterization of the Complex

Samples were characterized by infrared (IR) and photoluminescence (PL) spectroscopy. Infrared spectra were registered with a PerkinElmer Spectrum 100 FTIR Spectrometer (Beaconsfield, UK) with a resolution of 1 cm^−1^. IR spectra were recorded on the dry powder between KBr pellets or in Nujol mull between KBr pellets. Elemental analyses for C, H, and N were performed on an Elemental Analyzer system GmbH (Vario El cube, Langenselbold, Germany).

Commonly, PL emission spectra were recorded with a resolution of 0.0715 nm using different excitation sources with an MDR-23 single emission monochromator (LOMO).

High-resolution PL emission spectra (0.2 cm^−1^) were registered with a double grating spectrophotometer DFS-52 (LOMO) (St Petersburg, Russia) and a Hamamatsu photomultiplier module H8259-01 (Hamamatsu City, Japan) in a photon counting mode. PL spectra were registered with a Thorlabs LD (Newton New Jersey, USA) (CPS405 Collimated Laser Diode Module, 405 nm, 4.5 mW) as an excitation source. A pulsed nitrogen laser at 337 nm with a repetition rate of 10 Hz and a pulse width of 10 ns was used for the PL time decay measurements. The emitted light was detected with a module H8259-01 and a counting unit C8855-01 connected to a PC. The excitation spectra were registered with an MDR-23 monochromator as an excitation source and a double grating spectrometer DFS-52 for collecting the PL emission. A halogen lamp Osram (Munich, Germany) 64623 HLX 12V 100W was used as a light excitation source.

The PL time decay was recorded using a nitrogen pulsed laser at a repetition rate of 10 Hz and a 100 MHz digital storage oscilloscope (GW Instek, Taipei, Taiwan) (GDS-820 100 MHz) with a resolution of data acquisition of 4 µs. The measurement of the quantum yield was performed using the method of an integration sphere [27,28]. The PL temperature studies were carried out with a Leybold RDK 10-3202 closed-cycle refrigerator system (Vienna, Austria). The temperature of the samples was controlled by a thermocouple with an accuracy of 0.02 K. The emission spectra were corrected for the instrument spectral sensitivity.

## 3. Results and Discussion

### 3.1. Infrared Spectra

The IR spectrum of the complex [Eu(µ_2_-OC_2_H_5_)(btfa)(NO_3_)(phen)]_2_phen is presented in Appendix A (see Appendix A). The identification of characteristic absorption bands was carried out by comparison with the reference data [29,30]. The overall pattern of the IR spectrum reflects the molecular structure of the compound. The IR spectrum of o-phenanthroline monohydrate is characterized by the following absorption bands (ν, cm^−1^): 3369s (wide), ν(OH) 3061m, 2988m, 2902m, 2613w, 2184w, *1980w (wide), *1834w, *1768w, *1646m, 1617m, 1587m, 1562m, 1503s, 1493m,1447m, 1422s, 1406s, 1395m, 1346m, 1296w, 1250m, 1242m, 1232m, 1218m, 1137m, 1091m, 1079w, 1037w, 987w, 957w, 883m, 854vs, 778m, 739vs, 724m, 707s, 624s, 508w, and 411w (*overtone bands and component frequencies of δ(CH) non-planar oscillations of the aromatic ring in the region 1000–700 cm^−1^; w—weak; sh—shoulder; m—medium; s—strong; vs—very strong).

Absorption bands in the IR spectrum of btfa (ν, cm^−1^): 3676w, 3122w, 3070w, 2989m, 2973m, 2902m, 1908w, 1817w, 1600s, 1591s, 1576s, 1491m, 1473m, 1411w, 1394w, 1384w, 1342w, 1322w, 1254s, 1203vs, 1178s, 1141vs, 1117vs, 1098s, 1088sh, 1068s, 1029m, 997m, 974w, 936w, 896s, 847m, 814m, 804w, 773vs, 717m, 689vs, 677sh, 625s, 579s, 518w, 481w, and 440w.

Absorption bands in the IR spectrum of the complex [Eu(µ_2_-OC_2_H_5_)(btfa)(NO_3_)(phen)]_2_·phen (ν, cm^−1^): 3078w, 2985w, 2902w, 2613w, 2322w, 1637m, 1610s, 1598m, 1575s, 1541m, 1531m, 1520m, 1498w, 1489m, 1473m, 1459sh, 1441w, 1426m, 1377w, 1388w, 1318s, 1306m, 1290vs, 1250sh, 1241m, 1223w, 1180vs, 1135vs, 1106m, 1097w, 1078m, 1053w, 1038w, 1026m, 1002w, 968w, 944m, 896w, 864m, 847m, 808m, 796m, 767s, 731m, 718m, 713sh, 700s, 683m, 630s, 580s, 554w, 517m, 460m, 433w, and 418w.

Absorption bands registered in the IR spectra at 1180 and 1135 cm^−1^ are attributed to valence oscillations of the group CF_3_, ν_as_, and ν_s_ respectively. The registration of absorption bands at 731 cm^−1^ and 700 cm^−1^ (out-of-plane vibration δ(CH) in the aromatic ring of the substitution type in the benzene ring) indicates the presence of a 1-substituted benzene ring (or five adjacent hydrogen atoms) that confirms the presence of btfa in the europium(III) complex. Ethyl radicals (C_2_H_5_) were identified in the complex by absorption bands at 1459 cm^−1^ ν_as_(CH_2_/CH_3_) and 1377 cm^−1^ ν_s_(CH_2_/CH_3_), 1473 cm^−1^ (scissor oscillations of CH_2_), and 1470 cm^−1^ and 1466 cm^−1^ δ(CH_2_) [29]. Nitrate ions NO_3_^-^ are revealed by strong absorption bands at 1489 cm^−1^ and 1290 cm^−1^, as well as by a characteristic “breathing” band at 1026 cm^−1^ as the most likely coordinated metal in chelate-bidentate mode [29].

*o-Phenanthroline* in the complex was identified by absorption bands at 3061–2800 cm^−1^ ν(CH); 1637 cm^−1^ ν(C=N); and 1575, 1498, and 1441 cm^−1^ ν(C=C) in the aromatic ring, as well as by the presence of out-of-plane δ(CH) absorption bands, which characterize the type of substitution in the benzene ring at 847 cm^−1^ (1,2,3,4-substituted benzene ring or two adjacent hydrogen atoms) and 767 cm^−1^ (1,2,3-substituted benzene ring or three adjacent hydrogen atoms), characteristic for o-phen [29,30].

The presence of monoanions of benzoyl trifluoroacetone (btfa) in the complex is confirmed by the registration of a strong absorption band at 1610 cm^−1^. This band can be attributed to the carbonyl group being weakened by the resonance between the C–O–M and C=O→M bonds as a result of the formation of the pseudo-aromatic ring within the coordination of the btfa ligand to Eu(III) [29]. In the spectrum of btfa, the band ν(C=O) can be observed as a strong absorption at 1600 cm^−1^, which in the solid state contains intramolecular hydrogen bonds [31]. In the IR spectrum of the complex, this band is very intensive and shifted to a higher frequency at 1610 cm^−1^. The shift of this band in the complex and the appearance of a new band at 460 cm^−1^, with respect to the spectrum of the ligand, indicate the coordination of btfa to the europium(III) ion by means of oxygen atoms [23]. Other new bands that appear in the spectrum of the complex in the region 560–400 cm^−1^ compared to the spectra of o-phenanthroline and btfa, namely at 554, 433, and 418 cm^−1^, can be attributed to other Eu-O bonds with NO_3_^-^ ions and OC_2_H_5_ bridges.

The band ν(C=N) in the spectrum of o-phenanthroline monohydrate appears at 1646 cm^−1^, while in the spectrum of the complex, the band moves to a lower frequency of 1637 cm^−1^, which indirectly demonstrates the coordination of o-phenanthroline molecules to the europium(III) ion. The band at 1637 cm^−^^1^ is related to the ν(C=N) stretching frequency, the value being typical for the imine functional group coordinated to Ln(III) ions [32]. The coordination of this ligand to the metal atom is proved directly by the band of medium intensity at 517 cm^−1^ [23]. The IR absorption pattern is found to be in good agreement with the proposed molecule structure of the compound (Figure 1).

### 3.2. Photoluminescence and Discussion

The excitation spectrum of the complex (1) was recorded at 300 K by monitoring the emission at 612 nm, corresponding to the ^5^*D*_0_→^7^*F*_2_ transition of Eu^3^^+^ (Figure 2). The excitation spectra of the complex exhibit a broad band between 300 and 450 nm with a peak maximum at 376 nm, a shoulder at 335 nm, and a sharp peak at 468 nm. The sharp peak at 468 nm is ascribed to the intra-configurational 4*f*–4*f* transitions in Eu^3^^+^ ion, ^7^F_0_→^5^D_2_, along with a very weak peak at 395 nm, ascribed to ^7^F_0_→^5^L_6_ configurational transition [33,34,35]. The spectrum intense band with a peak maximum at 376 nm, along with the band with a shoulder at 335 nm, are related to the π-π* transitions of the ligands [33,36]. The excitation spectrum confirms the efficient blue-light sensitized luminescence of the complex (**1**). The energy transfer process, specifically in the case of blue-light excitation PL, is rather complex, involving several electronic states from both the ligands and the metal ion, as well as several different mechanisms, and has been largely discussed in the literature [24,25,35].

The PL emission spectra of powder samples of (**1**) were registered in the temperature range from 300 K to 10.7 K under excitation at 405 nm, which is close to the maximum absorption of [Eu(µ_2_-OC_2_H_5_)(btfa)(NO_3_)(phen)]_2_phen. The PL emission spectra exhibit (Figure 3) characteristic metal-centered luminescence bands, assigned to the internal radiative transitions of the Eu^3+^ ion, ^5^D_0_→^7^F_j_ (*j* = 0–3) and ^5^D_0_→^7^F_j_ (*j* = 0–4). The complex major emission bands are governed by the radiative transitions from the first excited ^5^D_0_ level to the ^7^F_j_ (j = 0–4) manifold, with the barycenters at 579.9 (^7^F_0_), 589.9 (^7^F_1_), 611.8 (^7^F_2_), 651.4 (^7^F_3_), and 704.7 nm (^7^F_4_), respectively.

The most intense transition, which dominates the luminescence spectrum of the complex, is the electric dipole transition ^5^D_0_→^7^F_2_ with a peak at around 612 nm. The hypersensitive to the site symmetry of the Eu^3+^ ion ^5^D_0_→^7^F_2_ transition exhibits a well-resolved fine structure, determined by the influence of the ligand’s molecular electric field on the degenerated Eu^3+^ ion level ^7^F_2_ [18,19,20].

Decreasing the temperature of the sample leads to decreasing the bandwidth and increasing the intensity of the emission peaks (Figure 3), which is due to the electron–phonon interaction with the luminescence center. However, in the case of the emission band ^5^D_0_→^7^F_0_, the peak intensity, in the low-resolution spectra, seems to not change with temperature (Figure 3 and Figure 4). Although, the FWHM of the transition ^5^D_0_→^7^F_0_ exhibits a clear tendency of narrowing (Appendix A) while cooling the sample down, similarly to all other transitions. The band ^5^D_0_→^7^F_0_ at~580 nm represents a quite notable feature in the luminescence spectrum of the [Eu(µ_2_-OC_2_H_5_)(btfa)(NO_3_)(phen)]_2_phen complex. It has a small line width, which at 300 K is 11.9 cm^−1^. The ^5^D_0_→^7^F_0_ transition is forbidden by the selection rules and its registration suggests that the europium ions are located in a low-symmetry environment [20,37].

In the case of low-resolution PL spectra (Figure 4), the ^5^D_0_→^7^F_0_ transition appears as a single, almost symmetrical emission band, whose peak position corresponds to 17,243.8 cm^−1^ at room temperature and to 17228 cm^−1^ at 10.7 K. The temperature shift of the ^5^D_0_→^7^F_0_ peak position in a low-resolution spectrum is represented in Figure 4 (the inset). The total magnitude of this shift (300–10.7 K) is 14.1 cm^−1^. Two different regions of behavior appear, well resolved at 90 K. The shape and relatively broad width of the transition ^5^D_0_→^7^F_0_ (FWHM equal to 11.9 cm^−1^ at 300 K) suggest that it may contain two closely spaced components. Indeed, a high-resolution spectrum of the transition (Figure 5) shows that it consists of two narrow (<6 cm^−1^) emission lines labeled A and B, as well as one very weak component, C. The separation of the two main components is only 0.96 meV. The main components A and B dominate the spectrum of the band ^5^D_0_→^7^F_0_ with more than 98% of the integrated intensity, while the weak component C represents about 1.8% of the integrated intensity of the transition.

Because of the negligible contribution of the C component, it will be ignored in the following analysis. Most probably [38,39], the low-intensity component C originates in a small number of Eu^3+^ ions incorporated in the matrix as defects. Since the ^5^D_0_→^7^F_0_ emission line cannot be split by the crystal field, the presence of two-component lines in the ^5^D_0_→^7^F_0_ spectrum clearly indicates that there are two different sites of the Eu^3+^ ion in the complex. The small split of about 1 meV between the two components A and B (Figure 5) suggests similar environments of both Eu^3+^ ions [38,39].

The registration of the ^5^D_0_→^7^F_0_ transition is associated with a low symmetry complex, containing the Eu^3+^ ions that most probably occupy a site with C_nv_, C_n_, or C_s_ symmetry, since other symmetries do not give an observable ^5^D_0_→^7^F_0_ transition [18,19,20,32,33]. However, it needs further confirmation by the X-ray diffraction (XRD) investigations of the compound. The constant intensity of the PL peaks in Figure 4 is probably determined by the low resolution of the spectrum. Because of this, it does not resolve the individual ultra-narrow components of the transition (Figure 5), which may vary in intensity while the sample is cooled down.

The emission band at 585–600 nm is related to magnetic dipole transitions ^5^D_0_→^7^F_1_, and it reflects the crystal-field splitting of the ^7^F_1_ level (Figure 6). In a low symmetry compound for a single Eu^3+^ site, the total removal of crystal field degeneracies results in three sublevels, the maximum number of 2j + l components for ^7^F_1_ [18,19]. In Figure 5, one can distinguish no less than six main components for the ^5^D_0_→^7^F_1_ transition, accompanied by a number of weak satellites. The splitting pattern of the ^5^D_0_→^7^F_1_ transition indicates the existence of at least two nonequivalent, nearly identical coordination environments of Eu^3+^ in the matrix [18,38]. As for the low-intensity satellite lines, we suppose that they may be related to vibronic transitions, defects, or impurities [38,39,40,41].

A number of very weak transitions from the higher excited state ^5^D_1_ can also be seen in the high-resolution spectrum represented in Figure 7 (the inset). These transitions are ^5^D_1_→^7^F_0_ (barycenter 526.9 nm), ^5^D_1_→^7^F_1_ (537.7 nm), ^5^D_1_→^7^F_2_ (560 nm), and ^5^D_1_→^7^F_3_ (583 nm). Transitions originating at the ^5^D_1_ level to the ^7^F_0–3_ levels are short-lived. These figures are consistent with the data from [42]. Intramolecular energy transfers from ligands to Eu^3+^ ion lead to a population of short-lived ^5^D_1_ levels and long-lived metastable 5D_0_ levels, giving rise to the Eu^3+^ emission to the ground multiplet ^7^F_j_ (j = 0–4) [18,38].

If the structural difference between two Eu^3+^ sites is small, the energy difference between the different peaks in the ^5^D_0_→^7^F_0_ transition is also small [19,40]. As discussed in [40], in the case of two distinct isomers in the crystal structure, this usually can result in quite a large energy difference between the transitions in the ^5^D_0_-^7^F_0_ region. The splitting character of the ^5^D_0_→^7^F_0_ transition suggests a dinuclear molecule and not distinct isomers since the splitting is as small as 0.96 meV (~6 cm^−1^). Such a small energy separation of two components of the ^5^D_0_→^7^F_0_ transition may indicate that both Eu^3+^ ions are located in similar environments, defined by the coordination sphere of six oxygen and two nitrogen atoms (Figure 1) [38].

The most intense emission band in the luminescence spectrum of the complex is the band related to the transition ^5^D_0_→^7^F_2_ with the barycenter at ca 612 nm (Figure 8). This transition, which is hypersensitive to the site symmetry of the Eu^3+^ ion, exhibits a fine structure, particularly well resolved at low temperatures.

For example, at 10.7 K one can distinguish the splitting of this transition into more than six components. This is another indication of the existence of two nonequivalent, nearly identical coordination environments of Eu^3+^ in the matrix. The hypersensitive transition ^5^D_0_→^7^F_2_ is much more intense compared to the magnetic dipole transition ^5^D_0_→^7^F_1_, and the ratio of the integrated intensities of two emission bands I(^5^D_0_-^7^F_2_)/I(^5^D_0_-^7^F_1_) (the asymmetry factor R) is ~9.02, which suggests that the Eu^3+^ ion is at a site without a center of inversion [20]. 

The ^5^D_0_→^7^F_2_ transition dominates the emission spectrum. Compared to other transitions, the much stronger intensity of ^5^D_0_→^7^F_2_ indicates that the ligand field surrounding the Eu^3+^ ion is highly polarizable and the Eu^3+^ ion is at a site without a center of inversion [18,19,23]. A number of very weak emission bands, associated with the ^5^D_0_→^7^F_3_ transition, can be observed in the range of 650–665 nm (Figure 9). This transition splits into eight crystal field components, which are best resolved at 10.7 K The ^5^D_0_→^7^F_3_ emission band is very weak, and this transition can only gain intensity via the j-mixing of states [43,44,45]. The PL emission peaks at 695–710 nm (Figure 10) can be assigned to the electric dipole transition ^5^D_0_→^7^F_4_. This transition is better resolved as the temperature decreases, and at 10.7 K, one can see it splitting into at least twelve peaks (for one Eu^3+^ site the maximum number of 2j + l components for ^7^F_4_ is nine).

While decreasing the temperature, in addition to a small temperature shift of the peaks, one can observe the PL emission lines narrowing. The temperature dependence of the integrated intensity for different PL emission bands ^5^D_0_→^7^F_j_ (j = 0–4) is illustrated in Figure 11. For all registered emission transitions ^5^D_0_→^7^F_0–4_, the integrated intensity seems to be constant, and no PL quenching is observed. Such behavior for the integrated intensity temperature dependence can be tentatively explained by the fact that the internal atomic transitions occurring in the 4f shell are not affected by temperature variation within the range of 10.7–300 K.

As the temperature is lowered from 300 K to 10.7 K, no differences in the relative ratio of intensity I(^5^D_0_-^7^F_2_)/I(^5^D_0_-^7^F_1_) can be observed, which suggests that no transformation in the structure occurs [46]. The luminescence decay profiles for compound **(1)** were monitored at 612 nm in powder samples under the excitation of laser pulses of 337 nm (Figure 12). Dots represent experimental data, and the blue lines represent the PL decay fitted by a two-exponential function:(1)$$I\left(t\right)=A_{1}\exp\left(-\frac{t}{\tau_{1}}\right)+A_{2}\exp\left(-\frac{t}{\tau_{2}}\right)$$ where $A_{1}$ and $A_{2}$ are pre-exponential factors, and $\tau_{1}$ and are the time constants. The time decay constants obtained from the plot in Figure 12 are as follows: $\tau_{1}$ = 0.67 ms, $\tau_{2}$ = 0.82 ms, $A_{1}$ = 1.0, $A_{2}$ = 0.787 respectively. The measured luminescence decay of the Eu(III) ^5^D_0_→^7^F_2_ transition is bi-exponential, suggesting the existence of two nonequivalent, nearly identical coordination environments of Eu^3+^ in the matrix.

The measured decay profile can be used for the calculation of the radiative lifetime ($\tau_{r}$) of Eu(^5^D_0_), and of the intrinsic quantum yield ($Q_{Eu}$). The observed luminescence lifetime $\tau_{obs}$ of the ^5^D_0_ excited state is assumed to be equal to the average lifetime [47], evaluated from the PL decay curve of the bi-exponential decay (Figure 12):(2)$$\tau_{av}=\frac{A_{1}\tau_{1}^{2}+A_{2}\tau_{2}^{2}\;}{A_{1}\tau_{1}+A_{2}\tau_{2}}$$

The average lifetime $\tau_{av}$, calculated from Equation (2), is found to be equal to 0.75 ms. The radiative lifetime $\tau_{r}$ was determined based on the measured PL emission spectrum, which was corrected for instrumental sensitivity. The radiative decay time $\tau_{r}$ for the Eu^3+^ ions was calculated following the limits of the Judd–Ofelt approach and the magnetic dipole nature of the ^5^D_0_→^7^F_1_ transition [18,32,43]:
(3)$$\frac{1}{\tau_{r}}=A_{MD,0}n^{3}(\frac{I_{T}}{I_{MD}})$$
where $A_{MD,0}$ is the probability of spontaneous emission for ^5^D_0_→^7^F_1_ transition with the constant value of 14.65 s^−1^; $I_{T}$ is the total integrated area of the PL spectrum; $I_{MD}$ is the integrated area of the magnetic dipole transition; and $I_{MD}$ is the refractive index of the matrix, assumed to be equal to 1.5 [18]. From Figure 2, $\frac{I_{T}}{I_{MD}}$= 14.95, and the radiative lifetime $\tau_{r}$ calculated from (3), is found to be 1.35 ms. The lifetime $\tau_{r}$ is related to the intrinsic quantum yield $Q_{Eu}$ and the observed luminescence lifetime $\tau_{obs}$ through the equation below [45]:(4)$$Q_{Eu}=\frac{\tau_{obs}}{\tau_{rs}}$$

The intrinsic quantum yield calculated from (4) is found to be 55.1%. The overall luminescence quantum yield Q of the complex is determined by the efficiency of the ligand-to-metal energy transfer $\eta_{sens\;}$ (sensitization efficiency) and the intrinsic quantum yield $Q_{Eu}$ through the equation below [45,48]:(5)$$Q=\eta_{sens\;}\times Q_{Eu}$$

The overall luminescence quantum yield Q of the complex measured by the integrating sphere was found to be equal to 49.2%. From Equation (5), the sensitization efficiency was found to be $\eta_{sens\;}$= 89.3 %. The quantum yield and emission lifetime of the synthesized complex (**1**) are comparable with others reported in the literature, particularly those containing the 1,10-phenanthroline ligand [11,33,35].

## 4. Conclusions

A new dinuclear coordination complex [Eu(µ_2_-OC_2_H_5_)(btfa)(NO_3_)(phen)]_2_phen was synthesized and characterized by IR and PL spectroscopy. PL emission spectra have been registered at different temperatures in the range of 300 K–10.7 K. The complex exhibits characteristic metal-centered luminescence bands assigned to the internal 4f radiative transitions of the Eu^3+^ ion, ^5^D_1_→^7^F_j_, and ^5^D_0_→^7^F_j_ (*j* = 0–4). The splitting pattern of the ^5^D_0_→^7^F_j_ (*j* = 0–4) transitions indicates that the europium ions are located in a low-symmetry environment. The high-resolution spectrum of the ^5^D_0_→^7^F_0_ transition shows that it consists of two narrow components, separated by 0.96 meV, which clearly indicates the presence in the matrix of two distinct, although chemically very similar, emitting Eu^3+^ sites. The excitation spectrum confirms the efficient blue-light sensitized luminescence of the complex. The absolute PL quantum yield, the intrinsic quantum yield, and the sensitization efficiency of ligands were determined to be 49.2%, 55.1%, and 89.3%, respectively. The complex shows potential applications in optoelectronics and biochemistry. Further efforts will be made to perform XRD studies to clarify the structure of the complex and its correlation with the PL emission spectra.

## 5. Patent MD 4677 B1 2020.02.29 (Republic of Moldova)

Europium(III) dinuclear coordinating compound with mixed ligands that shows luminescent properties. (Compus coordinativ dinuclear al europiului(III) cu liganzi micşti, care manifestă proprietăţi luminescente).

Int.Cl: *C09K 11/06* (2006.01), *C09K 11/77* (2006.01), *C07F 5/00* (2006.01), *C07C 49/92* (2006.01), *C07D 471/04* (2006.01).

Nr. deposit: a 2018 0063; Date: 2018.08.17; Date publ.: 2020.02.29, BOPI Nr. 2/2020.

Authors: Zubareva Vera, Bulhac Ion, Bordian Olga, Verlan Victor, Culeac Ion, Enachescu Marius, Moise Calin Constantin.

## Figures and Tables

**Figure 1 nanomaterials-12-02788-f001:**
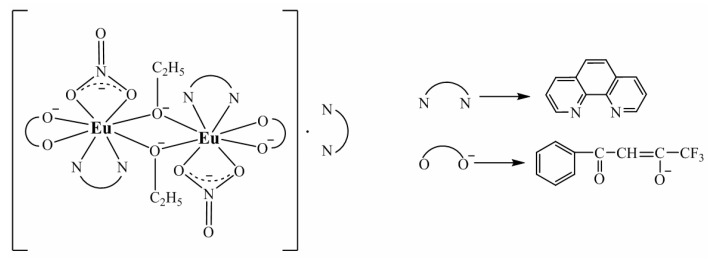
Proposed molecular structure of the complex [Eu(µ_2-_OC_2_H_5_)(btfa)(NO_3_)(phen)]_2_phen.

**Figure 2 nanomaterials-12-02788-f002:**
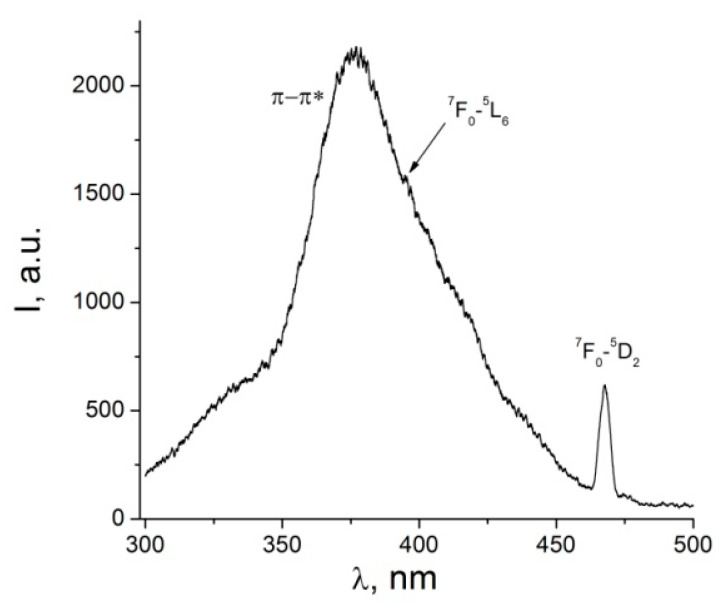
Low-resolution excitation spectrum of the complex (**1**) at 300 K for the ^5^D_0_→^7^F_2_ transition.

**Figure 3 nanomaterials-12-02788-f003:**
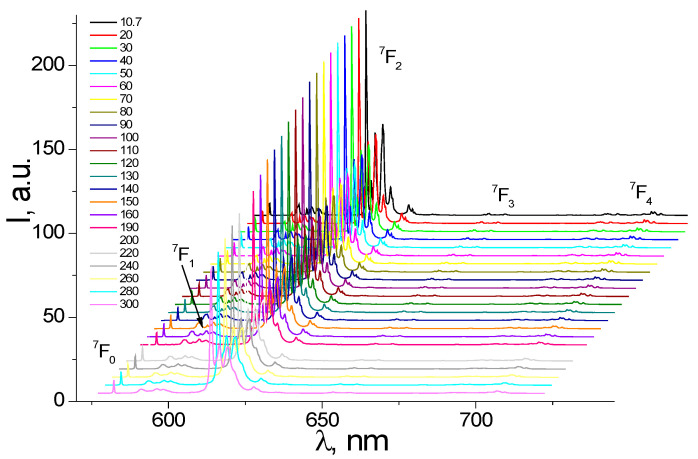
PL emission spectra of the powder sample measured at different temperatures (λ_ex_= 405 nm).

**Figure 4 nanomaterials-12-02788-f004:**
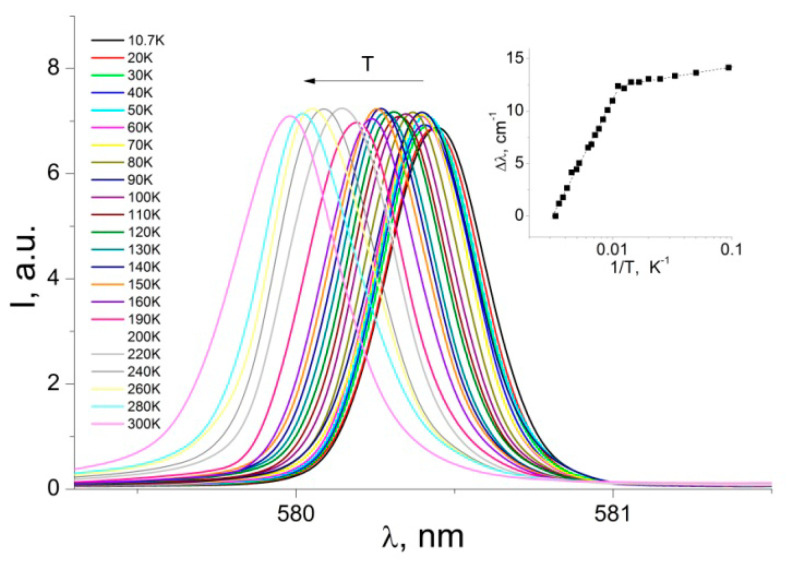
Low-resolution emission spectra for the ^5^D_0_→^7^F_0_ transition at different temperatures, 10.7–300 K, λ_exc_ = 405 nm.

**Figure 5 nanomaterials-12-02788-f005:**
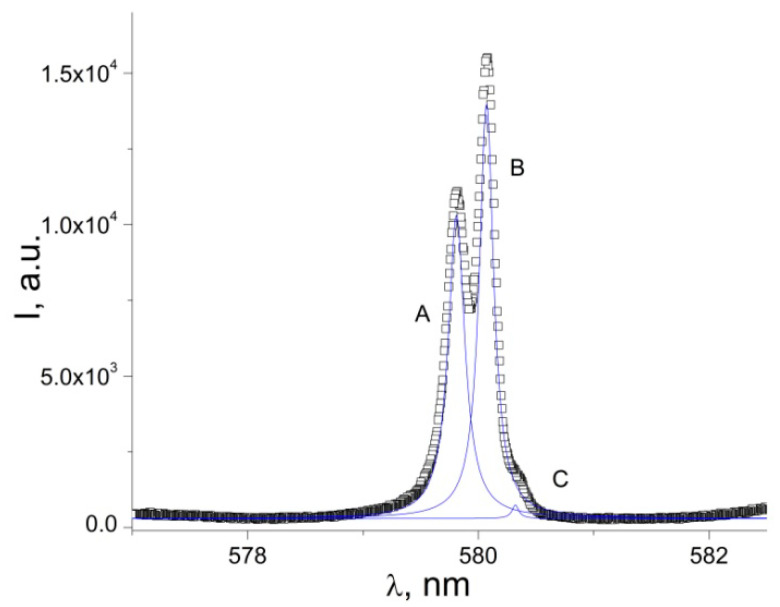
High-resolution emission spectrum for the ^5^D_0_→^7^F_0_ transition at 300 K and its deconvolution, λ_exc_ = 405 nm.

**Figure 6 nanomaterials-12-02788-f006:**
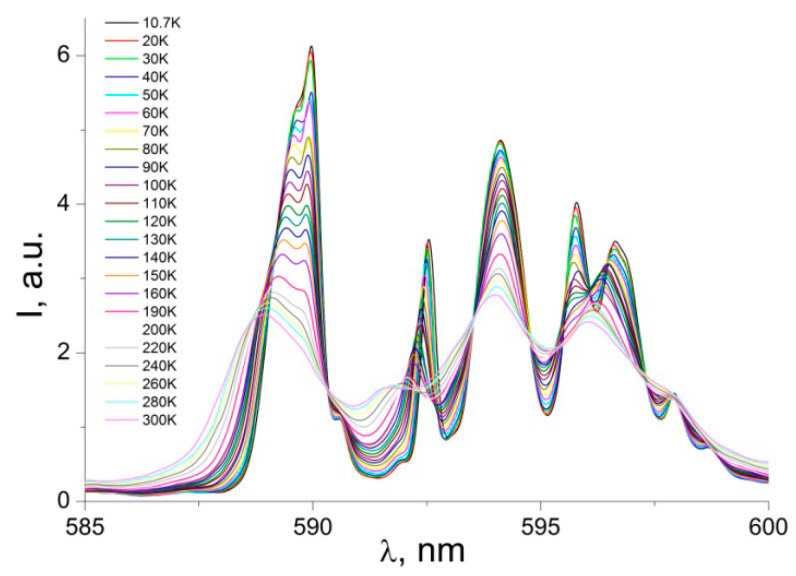
PL emission spectra of powder sample: magnetic dipole transition ^5^D_0_→^7^F_1_.

**Figure 7 nanomaterials-12-02788-f007:**
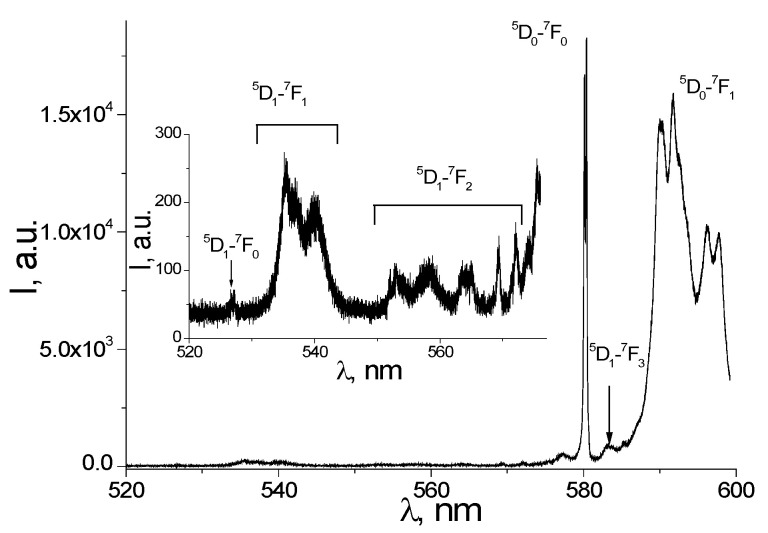
High-resolution emission spectrum for the 520–600 nm region at 300 K, λ_exc_ = 405 nm.

**Figure 8 nanomaterials-12-02788-f008:**
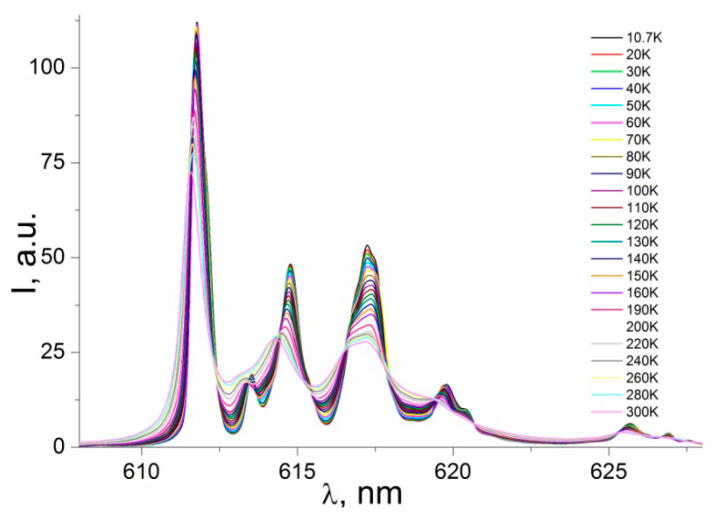
PL emission spectra of powder sample: electric dipole transition ^5^D_0_→^7^F_2._

**Figure 9 nanomaterials-12-02788-f009:**
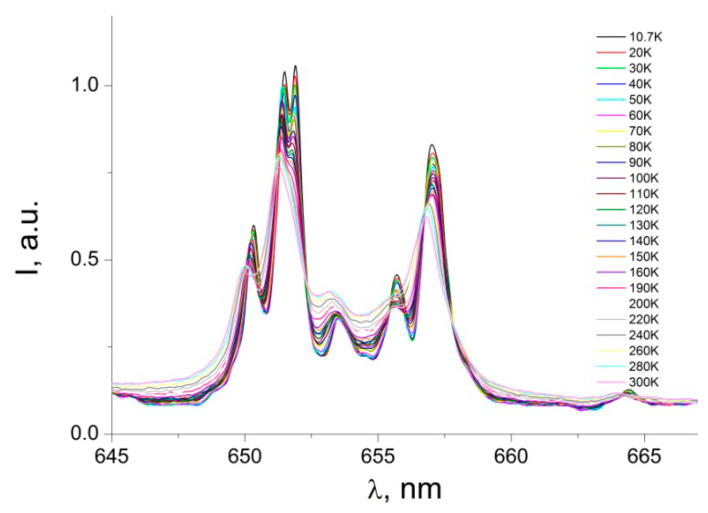
Photoluminescence emission spectra of the powder sample [Eu(µ_2_-OC_2_H_5_)(btfa)(NO_3_)(phen)]_2_phen complex: ^5^D_0_→^7^F_3_ transition.

**Figure 10 nanomaterials-12-02788-f010:**
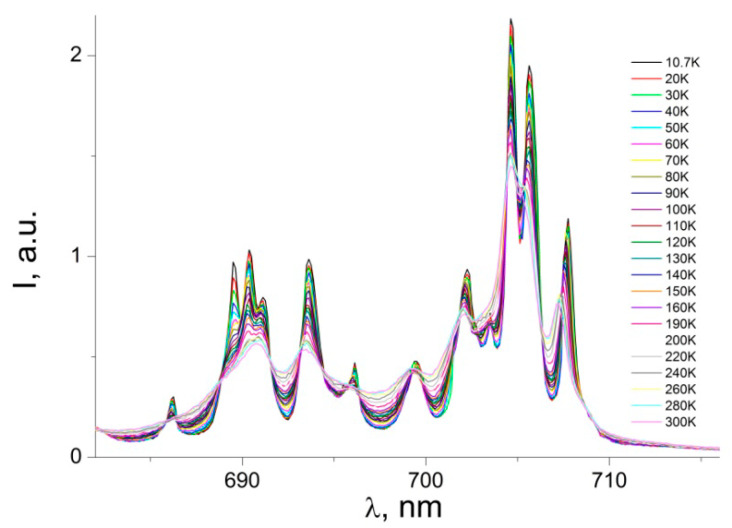
Photoluminescence emission spectra of the powder sample [Eu(µ_2_-OC_2_H_5_)(btfa)(NO_3_)(phen)]_2_phen complex: ^5^D_0_→^7^F_4_ transition.

**Figure 11 nanomaterials-12-02788-f011:**
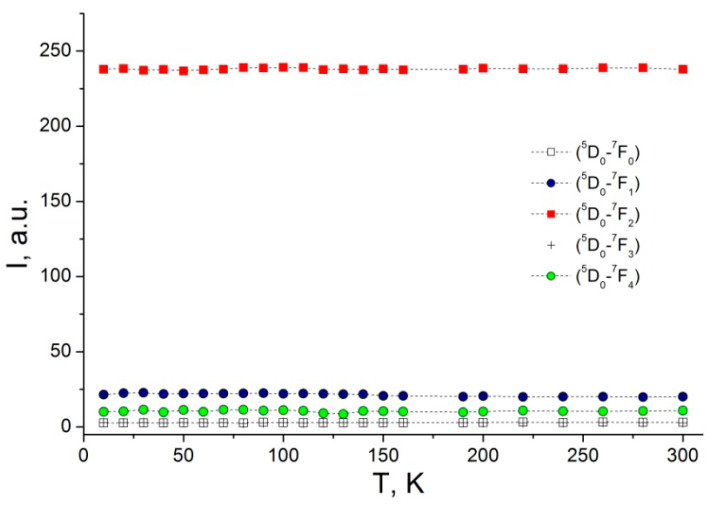
Temperature dependence of integrated emission intensity for different transitions ^5^D_0_→^7^F_j_ (j = 0–4).

**Figure 12 nanomaterials-12-02788-f012:**
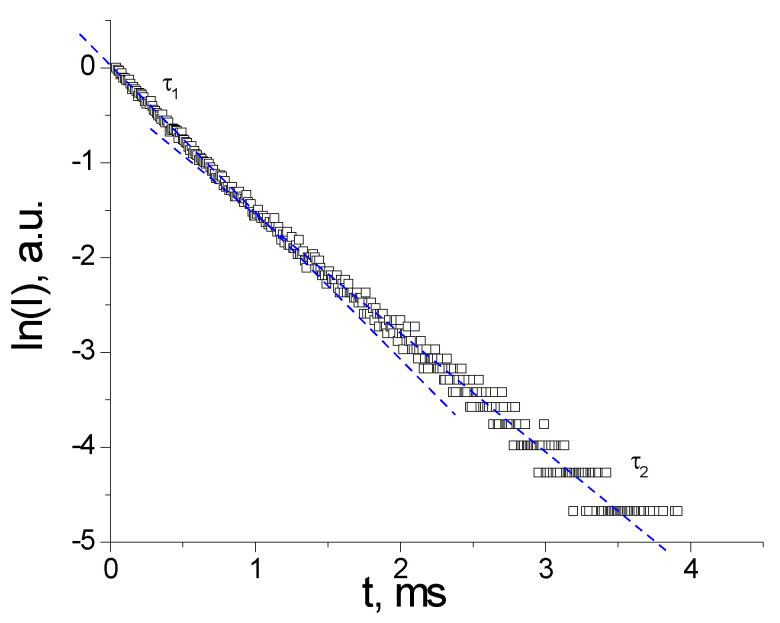
PL decay profile in powder sample at 300 K measured at 612 nm under pulsed excitation at 337 nm.

## Data Availability

Not applicable.

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
