# Peer review of "Synthesis and Characterization of Coordination Compound [Eu(µ2-OC2H5)(btfa)(NO3)(phen)]2phen with High Luminescence Efficiency"

_nanomaterials, 2022, doi:10.3390/nano12162788_

Round 1

Reviewer 1 Report

My comments are in attached file.

Author Response

Dear Reviewer,

We are very grateful indeed for your valuable comments and recommendations. Please find attached our responses.

Reviewer 2 Report

The article has reported on the synthesis and characterization of highly luminescent [Eu(μ2-OC2H5)(btfa)(NO3)(phen)]2phen compound. The compound was synthesized with a general organic chemistry approach at the author’s lab. The compound characterization was conducted with infrared and photoluminescence spectroscopies. The infrared spectroscopy was mainly used for the identification of organic and inorganic groups in the compound. The photoluminescence (PL) spectra were measured in the visible range under excitation at 405 nm in the temperature range from 10 to 300 K. The obtained PL spectra were associated with radiative transitions of Eu3+ ions. Likewise, the PL spectra indicated that Eu3+ ions were located in two non-equivalent low symmetry positions. The compound demonstrated high quantum yield of ~ 49.2 % elucidated from the life-time measurements. Broad fields of optoelectronics and biochemistry were outlined as possible applications.

I cannot recommend the article for publication in the present state. In my opinion, the article needs major revision due to the following reasons:

1.      Introduction section doesn’t provide sufficient information on the state-of-the-art synthesis of various lanthanide compounds and its particular applications. Optoelectronics and biochemistry are too broad fields nowadays. I urge authors to be more specific. Moreover, the majority of cited articles are older than five years. I’d recommend to the authors reading some new publications on lanthanide complexes and its modern applications, e.g.

        i.            Chem. Sci., 2021, 12, 271 6–2734, https://doi.org/10.1039/d0sc05419d

         ii.            Inorg. Chem. 2022, 61, 5972−5976, https://doi.org/10.1021/acs.inorgchem.2c00071

          iii.            Adv. Optical Mater. 2021, 9, 2101495, https://doi.org/10.1002/adom.202101495

      iv.            CrystEngComm, 2021, 23, 645-652, https://doi.org/10.1039/D0CE01477J

2.      Chemicals and materials section presents the description of the synthesis but it’s unclear if the authors developed the method themselves or they used an already existed approach. Likewise, the authors say that the compound is highly stable but they don’t show any supporting information, e.g., a series of photos over a long period of time.

3.      The measured infrared spectrum isn’t presented in the article. This is a missing experimental result.

4.      Figure 3 shows low-resolution emission spectra of the compound in the temperature range from 10 to 300 K. The intensity dependence is quite unusual. The intensity usually grows with cooling the sample down. Why the sample showed almost constant PL intensity over the wide temperature range?

5.      I encourage the authors to show the observed excitation and emission transitions of Eu3+ ion with Dieke diagram. That will significantly simplify understanding of PL mechanism. Likewise, it’s not clearly explained if Eu3+ ion directly excited by light, or its excitation occurs due to electron or energy transfer from the ligands.

Author Response

Dear Reviewer,

We are very grateful indeed for valuable comments and recommendations. Please find attached our responses.

Reviewer 3 Report

The synthesis of high-luminescent europium(III) coordination complex [Eu(µ2-OC2H5)(btfa)(NO3)(phen)]2phen is reported in this work. The structure of the complex was confirmed by IR spectroscopy and C/H/N Elemental Analysis. The high-resolution photoluminescence spectra of the complex were measured at different temperatures. Upon 405-nm excitation authors observed emission corresponding to the 5D1→7Fj and 5D0→7Fj (j = 0-4) transitions of the Eu3+ ion. The fine structure of the emission bands was carefully discussed and Eu3+ local symmetry was somehow proposed. The The absolute quantum yield and the sensitization efficiency were determined to be 49.2% and 89.3%. However, manuscript contains serious flows in data presentation and discussion. In principle, manuscript can be accepted after addressing the following comments:

1) The structure of the complex, especially the relative position of the ligands, is mostly speculated. I suggest to synthesize diamagnetic complex with yttrium or lanthanum, confirm their identical structure using X-Ray Powder Diffraction and measure 1H and 13C NMR spectra of Y/La complex to confirm the proposed structure. The synthesis is easy and suggested measurements should not take much time, but would be very useful to confirm the complex  structure. 

2) In “3.2. Photoluminescence emission characteristics” the authors discussed the fine structure of the emission bands and made an attempt to propose the local symmetry of the Eu3+ ion. However, the point symmetry group of Eu3+ ion was not clearly stated. I suggest to carefully analyse the fine structure of 5G0-7F2 hypersensitive emission transition and propose the possible group or several possible groups of symmetry of Eu3+ ion.

3) Lines 288-313: The quantum efficiency was calculated from the relative intensities of the emission bands and the emission time constant, and the quantum yield was measured by integrating sphere. The sensitization efficiency was calculated as a ratio of quantum yield and Eu3+ quantum efficiency. Thus, authors state that the synthesized complex demonstrates the antenna effect, and the Eu3+ emission is a result of sensitization. However, authors used 405-nm excitation wavelength, probably corresponding to the 7F0-5L6 f-f transition of the Eu3+. Or 405-nm promotes ligand into the pi-pi* excited state? I strongly suggest authors to provide the sensitization mechanism demonstrating possible energy transfer routes .

4) I suggest to provide excitation spectrum and discuss the nature of initially populated excited state upon 405-nm excitation.

5)  The Eu3+ antenna complexes are well-known compounds. I suggest to compare PL quantum yield and emission lifetime of the obtained complex with previously reported complexes, especially containing 1,10-phenantroline ligand.

Author Response

(The authors gave the same response as above.)

Reviewer 4 Report

This paper describes a new brightly emissive Eu(III) complex, [Eu(µ2-OC2H5)(btfa)(NO3)(phen)]2phen, prepared by treatment of Eu(NO3)3 with corresponding beta-diketone with presence of alkali, followed by adding phen ligand. The product prepared was adequately characterized and studied in terms of photophysics. At 300 K, its emit typical Eu(III) based luminescence with relatively high quantum efficiency. In general, the presented results can be published in this journal after taking into account the following minor points:

1. Synthesis of the Eu(III) complex should be somehow discussed in the paper. 

2. Please add the IR spectrum of the complex in Supporting Information file. 

3. Caption to Figure 1: replace “Illustration of molecular structure…” to “Proposed molecular structure…”. 

4. The citation list, in my opinion, is too short for this paper. I recommend extending it over the relevant works on previously reported emissive Eu(III) complexes, e.g. 10.1016/j.ica.2022.121007, 10.1016/j.jlumin.2022.118989, 10.1039/D1NJ02441H, 10.1134/S0022476621020116, 10.1007/s10895-005-2831-9, 10.1039/C8DT03621G. 

5. The sentence “Infrared spectra were recorded in the range 4000-400 cm-1 (suspensions in Nujol) for 92 the attenuated total reflection mode (ATR).” is more suitable for Experimental, rather than of the “Results and discussion” part.

6. If possible, please grow single crystals of the complex reported and perform X-ray structure determination.  

7. The mentioned preparative yield is better to specify with an accuracy of units. 

Author Response

(The authors gave the same response as above.)

Round 2

Reviewer 2 Report

The authors did a thorough revison of the manuscript. It seems that they partially addressed all my questions. The Dieke diagram for Eu3+ ion wasn't presented. Likewise, the peculiar temeperature dependece of photoluminescence in Fig. 4 isn't still explained. Along with the absence of change in intensity, the bands obviously don't get narrower with cooling down the sample. Why does that happen? Could the authors provide a table with intergrated intensities at the indicated temperatures for Fig. 4? I'm mostly concerned with Fig. 4 because all other figures show common temperature dependence with intensity growing upon cooling down the sample.

Author Response

Dear Reviewer,

Thank you for the revision and valuable comments indeed.

Please find attached the file with our responses. 

Thanks!

Reviewer 3 Report

Authors significantly improved the quality of the presentation, and now manuscript can be accepted in present form. 

Author Response

Many thanks for the revision and valuable suggestions!